# Nail Ultrasound in Psoriasis and Psoriatic Arthritis—A Narrative Review

**DOI:** 10.3390/diagnostics13132236

**Published:** 2023-06-30

**Authors:** Mihaela Agache, Claudiu C. Popescu, Luminița Enache, Bianca M. Dumitrescu, Cătălin Codreanu

**Affiliations:** 1Department of Internal Medicine and Rheumatology, “Carol Davila” University of Medicine and Pharmacy, 050474 Bucharest, Romania; mihaela.agache@reumatologiedrstoia.ro (M.A.); luminita.enache@reumatologiedrstoia.ro (L.E.); bianca.dumitrescu@reumatologiedrstoia.ro (B.M.D.); catalin.codreanu@reumatologiedrstoia.ro (C.C.); 2Clinical Center of Rheumatic Diseases, 030167 Bucharest, Romania

**Keywords:** ultrasound, psoriatic arthritis, psoriasis, nail enthesitis

## Abstract

Ultrasonography has advantages for assessing psoriatic arthritis (PsA) due to its ability to evaluate several targets, including joints, entheses, and tendons, but also skin and nails. Although ultrasound is widely used in PsA, nail ultrasound, despite its potential as a non-invasive method for the early detection of inflammation in the nail apparatus, has low applicability in medical practice, as probes with a higher frequency are needed compared with the frequency of probes usually used. In the present article, we have narratively evaluated the studies published in the last 5 years (19 February 2018–18 February 2023) on nail ultrasound value in the diagnosis and monitoring of PsA. The studies published during this period share common measurement parameters, such as the OMERACT definition of enthesis published in 2018. We included original articles published in PubMed and Web of Science using the following search terms in all possible combinations: psoriatic arthritis, psoriasis, ultrasound, and nail. Articles were declared relevant if they presented data on nail morphology, power Doppler evaluation and nearby structure evaluation, while focusing on digitorum extensor enthesitis. In most of the studies, common morphological parameters were generally increased in patients with psoriasis or PsA and were demonstrated to change with medication. The thickness of the extensor tendon was greater in patients with PsA and psoriasis versus controls and it was correlated with nail clinical changes, especially the presence of onycholysis. The presence of PD showed large variability in healthy individuals. The link between enthesitis and nail involvement is still a subject of controversy, taking into account the latest histological findings. The use of ultrasound in the evaluation of nail and DIP enthesitis remains a challenge and an area of research in the coming years.

## 1. Introduction

The prevalence of psoriatic arthritis (PsA) among psoriasis patients is 5–42% [1]. Approximately 10–55% of psoriasis patients experience nail changes, but the lifetime incidence of nail involvement can reach 90%. Only 5–10% of psoriasis patients have only nail involvement. In those with PsA, nail changes are more frequent, with a prevalence varying from 50 to 87% [2]. Nail clinical changes were initially described by Schons et al. [3] and include leukonychia, pitting, red spots, onycholysis, salmon spots, subungual hyperkeratosis and splinter hemorrhages. Most of the clinical changes are included in the NAPSI (Nail Psoriasis Severity Index) or modified NAPSI score [4].

Nail involvement is a known risk factor for the occurrence of PsA among psoriasis patients, especially its distal impairment type [5,6]. Nail psoriasis leads to a poor quality of life, with a 93% social disability with avoidance of social interactions and limiting daily and professional activities [7].

Although the relationship between the distal interphalangeal (DIP) joint with nail involvement was first described in 1994 [8], it was only about 15 years later that nail psoriasis began to attract more interest among dermatologists and rheumatologists (Figure 1 and Figure 2), especially since McGonagle et al. [9] hypothesized a connection between entheses, nails and DIP joints. 

McGonagle showed that the fibers of the extensor tendon of the fingers tightly link the periosteum of the distal phalanx with the nail bed and nail matrix. Thus, the nail was considered the link between the skin and the joints, and recently, it was integrated as part of the musculoskeletal system [10]. Apart from clinical examination, the psoriatic nail can be evaluated by dermatoscopy and ultrasound. A 2022 systematic review on nail imaging techniques in psoriasis patients revealed that ultrasound is the most widely used technique for assessing psoriatic nails, despite the fact that dermatoscopy has a lower cost and requires less training [11]. The main advantage of ultrasound over dermatoscopy is the ability of the former to visualize DIP joints and distal insertions of the finger extensor tendons.

Although ultrasound is widely used in musculoskeletal disorders, nail ultrasound, despite its potential as a non-invasive method for the early detection of inflammation in the nail apparatus, is rarely used in patients with PsA. The lack of a standardized ultrasound imaging technique is apparent in the cited literature as studies report using 15 MHz to 24 MHz ultrasound probes. This variability is mostly caused by cost and market availability, since 18–22 MHz probes are more widely available and purchased. Physically, the quality of nail strata images increases with probe frequency. Therefore, a probe frequency of 22–24 MHz should ideally be used in assessing nails in psoriasis and PsA. With the emergence of high-frequency ultrasounds on a large scale, nails began to be the focus of researchers. The normal ultrasound appearance of the nail is represented by a trilaminar structure with two hyperechoic parallel lines (dorsal plate and ventral plate) separated by a hypoechoic space. The ultrasound nail parameters evaluated in most studies are nail bed thickness (NBT) and nail matrix thickness (NMT). Spectral Doppler is used to calculate resistivity indices.

In the present article, we have narratively evaluated the studies published in the last 5 years (19 February 2018–18 February 2023) as a continuation of a review from 2018 that presented the value of nail ultrasound in the diagnosis and monitoring of PsA [12]. PUBMED and Web of Science were searched using combinations of the following keywords: “ultrasound” and “nail” with either “psoriatic”, “psoriasis” or “DIP”, revealing 106 non-duplicate titles, of which 46 were relevant to the present review and 22 studies were referenced for their original research content. A systematic review with a meta-analysis could not be performed on this study pool due to high design variability and small number of standard randomized intervention trials. Future development of the study niche could reveal meta-analytical content regarding ultrasound nail parameters and therapeutic nail response (see below). The studies published during this period share common measurement parameters, such as the OMERACT definition of enthesis published in 2018 [13]. We obtained the following variables from the studies: objectives, design, underlying disease, batch size, type of ultrasound, and nail parameters. We grouped the studies into different categories, namely, studies with a strict assessment of nail parameters, studies related to treatment sensitivity to change and the ability of ultrasound to capture this change, and studies that tried to demonstrate the relationship between nail and joint enthesis.

Generally, the weaknesses of the presented studies include insufficient data on confounders (e.g., professions involving repetitive and/or intense hand movement), the small number of patients enrolled in single centers, and the lack of control subgroups in therapeutic studies (with a few exceptions).

## 2. Assessing Nail Parameters

Most studies in which nail parameters are measured utilize the morphological description of nails as used by Worstman et al. [14] (Table 1), the Guttierrez et al. [15] power Doppler (PD) classification of the nail bed and nail matrix with a score between 0 and 3, and the OMERACT definition of enthesitis as applied to the digit’s extensor tendon from the distal phalanx [16]. The studies included patients with psoriasis, PsA and control groups who had nail impairment (at least in some studies and exclusively in others). The number of nails evaluated per patient was also different.

For example, Idolazzi et al. [17] characterized the nail changes in 82 patients with psoriasis and PsA and 50 control patients as assessed by ultrasound of the nail from the second finger of the dominant hand, which is considered by the same authors in previous studies to be the most significantly affected nail. Nail abnormalities were detected during clinical examination in 40% of the psoriasis patients and in 62.7% of the PsA patients. In those with a normal nail clinical examination, ultrasound abnormalities of the trilaminar structure were detected (of any degree in 3% of psoriasis patients and in 9% of PsA patients). Nail plate thickness (NPT) and NBT were significantly lower in the control group compared with patients with psoriasis or PsA. PD ultrasound was higher in the nail bed and entheses in patients with psoriasis or PsA versus the control group. An NPT of 0.63 mm can distinguish patients from healthy individuals, and this marker has a sensitivity of 70% and a specificity of 78%. No significant differences of nail ultrasound measurements between PsA patients and psoriasis patients were found.

Krajewska-Włodarczyk et al. [18] analyzed 69 patients (38 with psoriasis, 31 with PsA and the remaining 30 in the control group), each having a nail change in at least one finger. In all patients, NPT correlated with the duration of psoriasis, regardless of the presence of nail changes. NBT in patients with PsA correlated with the duration of arthritis. Similar to the previously described study, an increase in NPT and NBT was detected in both groups when compared with the control, even in fingers with no nail impairment. The most frequent Worstman changes in gray scale (GS) were type 1 (in 86% of psoriasis patients) and type 2 (in 77% of PsA patients). An increased PD signal in the extensor tendon was observed in patients with PsA compared with those with psoriasis. Within groups, the Doppler signal was statistically different between those with and without nail impairment. The thickness of the extensor tendon and other enthesopathy changes (e.g., loss of the fibrillar architecture, enthesophytes, bone production) were observed more frequently in psoriasis or PsA patients with nail changes, but statistical significance was reached only in patients with psoriasis. 

The observational study conducted by Mondal et al. [19] on 45 patients with PsA and 45 control patients, revealed type 2 morphological changes, as described by Worstman, as the most frequent (51.8%). Approximately 88% of the nails had structural ultrasound changes, the right first finger being the most frequently impaired. Even in clinically normal nails, there were ultrasound changes in 75.3% of cases. According to previous studies, NBT and NMT were significantly increased in patients with PsA, while NMT showed a positive correlation with NAPSI in patients with PsA.

In a study conducted by Mahmoud et al. [20], type 2 morphology (loosening of the borders of the ventral plate) was the only aspect observed in HC versus patients with PsA, suggesting that type 2 is not specific for the psoriatic nail. The measurements of the nail plate thickness were similar between PsA patients and HC, as previously reported [19].

Naredo et al. [21] assessed a large group of 60 PsA patients (21 patients with psoriasis and 20 control patients) and reiterated the fact that B-Mode changes (evaluated with a 22 MHz probe) are different between patients with psoriasis, PsA and control patients. These changes were significantly increased, even in psoriasis or PsA patients without clinical changes. In addition to Worstman scores of 2 or 3, the loss of ventral plate integrity and the involvement of both nail plates were exclusively found in patients with PsA or psoriasis, respectively. Color Doppler (CD) was included in the description of two scores which were developed by this group of researchers, namely, CDsc1 (the nail bed occupied by Doppler along a scale of 0–2) and CDsc2 (the contact of the Doppler signal with the ventral plate along a scale of 0–2). These scores did not detect a difference between psoriasis or PsA patients with or without clinical involvement and they could not distinguish between control nails and nails with psoriasis. The authors explained that the blood flow in the nail bed and the nail matrix is variable in healthy patients.

Aydin et al. [22] also found a higher PD score in healthy patients. NAPSI correlated with the B-mode evaluation, but apparently, the ultrasound parameters were independent from other skin or joint scores. In a second study [23], the authors also noted more vascularity in HC nails than in patients, which was possibly linked with the local pressure caused by inflamed structures in PsA. Kassman et al. [24] introduced two new terms. The first is NPI (nail plate impairment) with ranges between 1 and 12 (the number of nails affected by structural changes) with 10 in fingers and 1 in each metatarsophalangeal (MTP) joint bilaterally. The second term is NT (nail thickness), which addresses the difficulty of measuring NPT in cases of fusion. In such cases, a global measurement can be performed (at 6 mm away from the nail fold). In total, 64 patients with PsA and 26 controls were evaluated. An NUSG (nail ultrasonography score) index was calculated, including NPI, NT and DA (a Doppler activity range of 0–36), as a feasible, reliable (with an intra- and inter-agreement study) and discriminating method to predict PsA. The best nail for measuring NPI is the left thumb nail. The most significant nail regarding NBT and NT is the left fifth finger nail, while for DA, the best is the right third finger nail. The evaluation takes 6 min. The authors reported that 5 of the 312 evaluated nails revealed fusion. NPI > 1.5, NT > 27 and DA > 1.5 are good predictors of PsA. There was a relationship between NAPSI, the nail clinical score, and NUSG. The evaluation of the 12 nails, including the MTP1 bilaterally, was superior to the assessment of just one nail performed in previous studies. The authors concluded that no nail selection should be performed when diagnosing.

Considering healthy individuals, Bellinato et al. [25] very recently conducted a study on 27 healthy subjects in order to establish a nail scanning protocol. These authors observed a significant trend of decreasing NPT from finger 1 to finger 5. The Doppler signal varied in terms of degrees, from absent to very obvious. There were no differences related to the dominant hand. A proper scan protocol must cover all nails during their longitudinal and transverse evaluation.

De Rossi et al. [26] evaluated 35 patients with psoriasis, 31 with PsA and 35 controls, using an 8 MHz power Doppler probe. The nail matrix resistance index, NMRI, and the nail bed resistance index, NBRI, were calculated on fingers 2 and 3 bilaterally. NMRI and NBRI were not different between groups, and neither the degree of PD, nor NPT were higher in the groups with PsA and psoriasis compared with the control group. The morphological and functional features of the trilaminate structure were type 1 and 4 in patients with psoriasis and type 2 and 3 in patients with PsA. Only the latter were observed to be different when compared to the control group. As in other studies, the PD was not able to distinguish the blood flow, which has a large variability in the control group.

Idolazzi et al. [27] published a study which is similar to another from 2018, but which included patients with psoriasis, PsA, rheumatoid arthritis (RA) and osteoarthritis. NBT (assessed on the second finger) in groups with PsA and psoriasis was different compared with RA and osteoarthritis groups. The enthesis PDUS showed a significant difference between patients with PsA and the other groups. The PD of the nail bed in psoriasis and PsA patients was not different compared with the osteoarthritis and RA groups. 

## 3. Evaluation of the Nail and Enthesis Treatment Response

In a prospective study on a group of 41 patients with nail psoriasis affecting at least three fingers and a control group, Krajewska-Włodarczyk et al. [28] evaluated the ultrasound changes after treatment with acitretin for 6 months. The nails and the extensor tendon of the fingers were examined with a 24 MHz probe. Nail alterations in terms of NT, NBT, NMT, and DETT (digital extensor tendon thickness) were examined. The patients were divided into a subgroup with signs and a subgroup without signs of enthesopathy at the DIP level. As expected, in patients with psoriasis, significant increases in the thickness of the nail plates, nail bed and nail matrix were detected versus controls, and all were correlated with mNAPSI. There were no differences regarding laboratory parameters, psoriasis duration or the mNAPSI index between the two subgroups with or without enthesopathy (as defined by OMERACT). There were no differences in terms of matrix vascularization, but a more intense nail bed PD signal was observed in the enthesitis patients. Tendon thickness was also correlated with nail bed thickness, while the occurrence of enthesopathy and PD at this level was more frequent in fingers with nail damage. Following treatment with acitretin, a decrease in the thickness of the nail bed and the nail matrix was observed. The nail clinical changes correlated neither with the thickness of the extensor tendon, nor with the PD reduction at this level. Notably, acitretin seems to be ineffective at the enthesis level, and the improvement in psoriatic onychopathy observed with ultrasound after acitretin therapy was not accompanied by an improvement in the joint component in PsA. 

Comparatively, the same team of authors [29] performed a 6-month prospective study with methotrexate 15–25 mg/week on 32 patients (19 with psoriasis and 12 with PsA) with nail damage and extensor enthesopathy in at least one finger. Following treatment, NMT, NPT and also the thickness of the extensor tendon of the fingers decreased. Unlike acitretin, methotrexate induced simultaneous ultrasound improvement in psoriatic onychopathy and the PsA joint/enthesitis component. At baseline, patients with PsA had significantly increased dimensions of the extensor tendon and the nail bed, with no differences in terms of NPT and NMT. Type 1 Wortsman changes were more frequent in patients with psoriasis. The mNAPSI was correlated with tendon thickness, and in both groups, it was correlated with nail bed thickness, while in patients with psoriasis as well as those with PsA, enthesopathy occurred more frequently in fingers with nail damage. Methotrexate decreased NB, NM and NP thickness. In patients with psoriasis, the thickness of the extensor tendon decreased. In both groups, the PD intensity in entheses decreased, proving the effectiveness of methotrexate in entheses, but there was an enhanced nail signal in both groups.

Recent studies [30] show that treatment with apremilast for 52 weeks in psoriasis patients with nail damage improved the structural ultrasound appearance of the nail. Subcutaneous secukinumab in PsA patients [31] also reduced mNAPSI, a slower radiological progression being observed in patients with concomitant nail psoriasis.

Clinical practice can benefit from such studies. For patients with nail psoriasis and DIP pain without joint swelling, ultrasound detects subclinical changes in PsA (which allows treatment adjustment) and differentiates them from concomitant osteoarthritis. For the most common measurements, the 18 MHz probe is frequently found in ultrasound laboratories, and no further hardware requirements are needed. Therefore, monitoring nail ultrasonographic parameters is readily available and can be a useful tool in monitoring PsA treatment [31]. The advent of new biological and targeted synthetic drugs that can specifically target GRAPPA domains of PsA (the nail and the enthesis) suggests that the evaluation of their effectiveness should not be based only on clinical examination, but should also include precisely quantified imaging methods. Studies in the literature demonstrate their ability to assess ultrasonographic parameters on nails and entheses or only on nail parameters. The literature also contains mixed results, with both positive and negative reports regarding morphology and response to treatment, which leaves room for larger and better controlled studies.

## 4. The Connection between DIP Erosion and Nails

Antony et al. [32] explored the connection between nails and DIP joints by also assessing non-DIP joints (proximal interphalangeal or metacarpophalangeal joints). The DIP evaluation was radiological and included erosion, osteoproliferation and joint space narrowing. In a group of 134 patients, the presence of a form of nail dystrophy (especially in patients with onycholysis) was associated with erosion in the DIP joints of the corresponding finger, but not in non-DIP joints. Hyperkeratosis was associated with space narrowing, erosion in both DIP and non-DIP joints (but with a greater magnitude in DIP joints) and osteoproliferation. Nail pitting was not associated with erosion or osteoproliferation. The evaluation was performed on individual fingers with validated radiological methods. This study showed the radiological and clinical anatomical associations between nail dystrophy and DIP damage and reveals the need to examine the nail–enthesis–DIP complex during clinical practice.

Villani et al. [33] compared the extensor tendon thickness, as evaluated by ultrasound, with the presence of DIP erosions, as evaluated by high resolution peripheral quantitative CT of the DIP joint. The groups were as follows: patients with psoriatic onycholysis without PsA, patients with skin psoriasis, patients with PsA for reference, and a control group. The tendon thickness was greater in patients with onycholysis than in patients with psoriasis, and the association was more frequent in this group compared with patients with psoriasis. The authors concluded that onycholysis was associated with enthesopathy and erosion and it could be considered as a severity marker of psoriasis.

## 5. The Connection between Nails and Enthesitis

Most studies published on this topic are related to the causal link between enthesitis and nail damage (Table 2). Aiming to observe which type of nail damage is related to subclinical enthesopathy, Moya Alvarado et al. [34] evaluated NPT, NBT, PD with a 22 MHz probe and with 500 MHz PRF (pulse repetition frequency) in the five nails of the dominant hand in 48 patients with psoriasis and clinical remission and 23 patients with inactive PsA. Approximately 68.8% of patients had extensor finger enthesopathy, while NAPSI increased in those with subclinical enthesopathy from both groups. In a finger-by-finger analysis, enthesopathy occurred most frequently in fingers 1 and 2. No correlations were found between the presence of enthesopathy and the nail ultrasound indices.

The fact that nail damage is a risk factor for the development of PsA and especially for the DIP form, was also verified by Krajewska-Włodarczyk et al. [35]. They enrolled 72 patients with nail psoriasis (41 with psoriasis and 31 with PsA) and 30 in the control group, who were evaluated by mNAPSI, PASI and DAS66/68. The nails were measured with ultrasound in terms of NT/NPT, NBT/NMT and their morphology was evaluated according to the Wortsman classification as well as the extensor tendon thickness (as defined by OMERACT), while DIP joints were evaluated with a 24 MHz probe. NT/NPT and NBT/NMT as well as the extensor tendon thickness were all higher in patients with PsA and psoriasis compared with the control group. Morphologically, type 1 prevailed in patients with psoriasis, while type 2 prevailed in PsA patients. In both patient groups, enthesopathy (loss of architecture, enthesophytes, bone changes) and PD tendon vascularization were more frequent in fingers which had nail damage and these were clearly more frequent in patients with PsA than in those with psoriasis.

Mahmoud et al. [37] performed an ultrasound assessment of fingernails that included a study of morphological changes, measurement of NBT, NPT and adjacent skin (ST), and the extensor tendon thickness in a group of 33 patients with PsA. Ultrasound nail changes were associated with thicker extensor tendon and more erosions.

Elliot et al. [36], studying a group of 45 patients with PsA and nail involvement, using a 5–18 MHz probe and examining for PD with 6.6 and 8.8 MHz, confirmed what previous researchers have found: a correlation between the clinical and ultrasound nail involvement with DIP enthesitis (DIPUS). The paper also revealed a significant correlation between the ultrasound changes of the nails and MASEI (Madrid Sonographic Enthesitis Index), a clinical score of peripheral entheses

Regarding the ability to distinguish between psoriasis, PsA and OA, in a prospective cross-sectional study [39], 50 patients with PsA with nail and DIP involvement, 12 patients with psoriasis and nail involvement and 13 patients with osteoarthritis (OA) were evaluated radiologically, by ultrasound and MRI for new bone formation (NBF). The DIP joints 2–5 from the dominant hand were evaluated with a 15 MHz probe. The study reported that NBF on ultrasound and MRI was a marker for osteoarthritis, indicating reduced likelihood of PsA. No radiological, ultrasonographic or MRI findings were found to differentiate PsA from psoriasis.

Recently, Huang et al. [38] studied a group of 154 patients with PsA and 35 patients with psoriasis and connected nail damage with enthesis damage, as quantified by NAPSI scores, structural changes, the Wortsman classification, NBT, NMT and GUESS (Glasgow Ultrasound Enthesitis Scoring System), using a 6–18 MHz probe. There were no nail morphometric differences between patients with psoriasis and patients with PsA. The NAPSI and GUESS scores were significantly associated with nail thickness, independently of age, sex or BMI. The GUESS scores were higher in PsA patients. The distal patellar ligament and the Achilles tendon were significantly thicker in PsA patients.

Last year, Ruscitti et al. [40] published the results of a study on 59 cases of psoriasis, PsA with psoriasis and PsA without psoriasis, focusing again on nails (BUNES—Brown University Nail Enthesis Scale score) and entheses (global MASEI score), using a 27 MHz probe. In contrast to other studies, Ruscitti et al. enrolled patients without immunosuppressive treatments at the time of evaluation and patients who followed a washout period. There were no significant BUNES differences between groups. As many as 47% of psoriasis patients had enthesitis, and 38% of the patients with PsA without psoriasis had nail damage.

In all of the above studies, researchers aimed to establish imagistic correlations between nails and enthesis in the absence of histological studies. Histological studies are lacking in this field in recent years. As a consequence, there are no estimates of sensitivity and specificity of nail ultrasound compared with nail biopsy, which is the gold standard for diagnosis of nail onychopathy in psoriasis/PsA. Nail biopsy is rarely mandatory, unless nail involvement is still unclear after non-invasive measures (clinical aspect, dermatoscopy, ultrasound). Histological evaluation of DIP and enthesis at this level is performed by lateral and longitudinal biopsy with an indirect evaluation of the enthesis. The procedure requires expertise in order to avoid diagnostic errors. 

Initially, in 2009, McGonagle described the lamina of the extensor tendon of the fingers that connects with the nail, forming the so-called enthesis organ. In very recently published studies, Perrin et al. [41] found enthesitis of the extensor tendon of the fingers in patients with nail psoriasis and PsA involving DIP joints with lymphocytic and giant infiltrates at this level, but they concluded that there is no inflammatory flow from the enthesis to the nail or from the nail to the enthesis. One of the explanations could be that the so-called superficial lamina corresponds to a fibroelastic structure of the proximal nail fold and the fibrous root of the nail which separates the nail from extensor enthesis. However, imaging did not find this thin fan-like lamina in the longitudinal lateral section of the nail. The controversial conclusion of the study is that the nail affected by psoriasis remains micro-anatomically independent of the enthesis and DIP. Instead, this study demonstrated an infiltration with CD4 and CD8 T lymphocytes in equal numbers at the level of the nail dermis, an infiltration similar to that of synovial damage and different from the psoriatic skin lesions where CD4 T lymphocytes predominate. This raises the possibility that the connection between the enthesis and the nail is immunological.

## 6. Conclusions

As evident from the above cited studies, which used common parameters (NBT, NPT, NMT), there are morphological changes which are predominantly described in patients with psoriasis or PsA. (Some studies even propose cut-off values.) Their ability to change with medication was also demonstrated, which further highlights the need to evaluate and increase the accuracy to detect/measure these parameters. Related to the morphological changes on the Wortstman scale, type 1 prevails in patients with psoriasis, while type 2 prevails in PsA patients. As expected, the thickness of the extensor entheses is greater in patients with PsA, but it was correlated with nail morphology, especially with the presence of onycholysis or hyperkeratosis. The presence of PD in the nails tended to diverge slightly between studies, showing that this is variable in healthy individuals as well. However, study designs also share some limitations, such as the lack of standardization (including the use of gel on nails), the lack of nail and finger involvement prevalence estimates, the lack of susceptibility to change estimation, and the reproducibility of using gel on nails. In addition to diagnosis and monitoring, ultrasound changes can also help with prognosis.

Certainly, ultrasonography and histology studies will be able to demonstrate the sensitivity and specificity of the imaging method. Ultrasound could also become the main imaging tool for dermatology nail evaluation, both for inflammatory conditions such as psoriasis and PsA, but also to perform differential diagnoses with certain benign or malignant tumors or with various subungual collections. Ultrasound can also guide nail biopsy. The methodology for quantifying the Doppler signal at the periungual level in PsA can also enter the research agenda, considering that it has variability in healthy controls. Even though dispose of a core domain set published by GRAPPA that includes entheses and nails [42], we do not yet have standardized methods for measuring the nail parameters or the distal entheses. The development of these standards would enable comparison of therapeutic studies on the core set of domains.

## Figures and Tables

**Figure 1 diagnostics-13-02236-f001:**
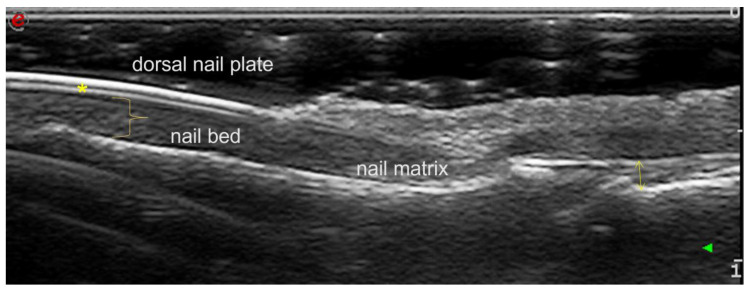
Normal nail examination in gray scale longitudinal scanning of the dorsal side (Esaote Mylab Twice device; 18 MHz linear transducer). Normal trilaminar appearance of the nail plate is observed, with homogeneous height (asterisk). The nail bed is a hypoechoic structure between the ventral plate and the periosteum of the distal phalanx (nail bed thickness: close brace). A normal thickness of the extensor tendon is observed (double arrow).

**Figure 2 diagnostics-13-02236-f002:**
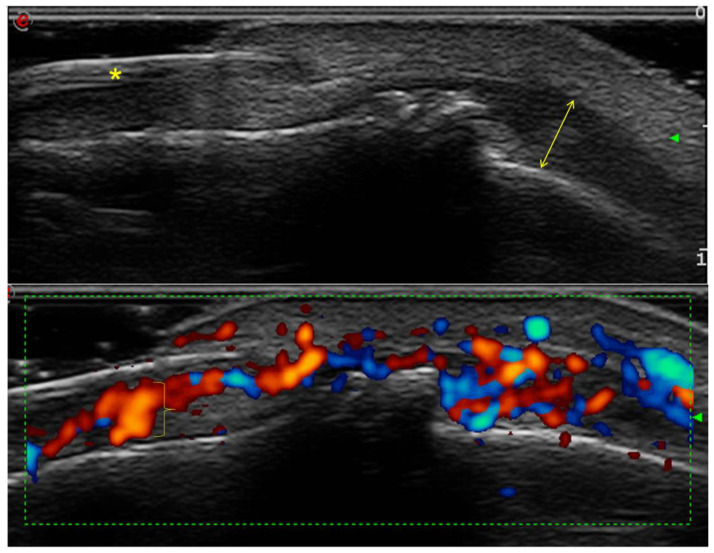
Psoriatic arthritis: examination in gray scale and power Doppler, with longitudinal scanning of the dorsal face—nail damage and enthesitis at the level of the second right DIP joint (Esaote Mylab Twice machine; 18 MHz linear transducer). The loss of the trilaminar appearance of the nail plate is observed (asterisk), appearing as a single hyperechoic layer, with inhomogeneous height. An increase in the thickness of the extensor tendon is observed (double arrow). Power Doppler detects increased flow in the nail bed (close brace) and at the level of the extensor tendon.

**Table 1 diagnostics-13-02236-t001:** Worstman’s morphological classification of ultrasonographic changes in the psoriatic nail [14].

type I	Intact and hyperechoic dorsal plate with focal hyperechoic areas on the ventral plate
type II	A normal hyperechoic dorsal plate with blurring and loss of ventral plate margins
type III	Wavy appearance of both nail plates
type IV	Loss of definition of both plates

**Table 2 diagnostics-13-02236-t002:** Summary of study results on nail US changes and enthesitis.

Study	Subjects	Probe (MHz)	Correlation between Nail US Changes (Morphological Parameters) and DIP US (Thickness of Extensor Tendon)	Correlation between Nail US Changes (Morphological Parameters) and Peripheral Enthesitis Score
Moya Alvarado et al., 2018 [34]	48 PsO 23 PsA	18–22	Not correlated significantly	-
Krajewska-Włodarczyk et al., 2019 [35]	41 PsO 31 PsA 30 HC	24	PsO: extensor tendon thickness correlated with NBT (r = 0.316, *p* = 0.027) and NMT (r = 0.421, *p* = 0.012); PsA: extensor tendon thickness correlated with NBT (r = 0.402, *p* = 0.031)	-
Elliot et al., 2021 [36]	46 PsA	18	Correlation between the nail US score and DIP US (r = 0.43, *p* = 0.003)	Correlation between nail US score and the active peripheral enthesitis score (MASEI-active; r = 0.35, *p* = 0.018)
Mahmoud et al., 2022 [37]	31 PsA	NS	Finger extensor tendon thickness correlated with NBT (r = 0.412, *p* = 0.00), NPT (r = 0.310, *p* = 0.00) and the thickness of the adjacent skin (r = 0.509, *p* = 0.00)	-
Huang et al., 2022 [38]	35 PsO 154 PsA	18	-	GUESS scores were significantly associated with nail thickness (*p* = 0.020)

DIP—distal interphalangeal joint; GUESS—Glasgow Ultrasound Enthesitis Scoring System; HC—healthy controls; PsA—psoriatic arthritis; MASEI—Madrid Sonographic Enthesis Index; NBT—nail bed thickness; NMT—nail matrix thickness; NPT—nail plate thickness; NS—not specified; PsO—psoriasis; US—ultrasound.

## Data Availability

Not applicable.

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
