# Peer review of "Nail Ultrasound in Psoriasis and Psoriatic Arthritis—A Narrative Review"

_diagnostics, 2023, doi:10.3390/diagnostics13132236_

Round 1
Reviewer 1 Report
The authors present a review of studies published in the past 5 years on psoriatic nail ultrasound and its value for the diagnosis and follow-up of psoriatic arthritis (PsA) during treatment. It is concluded that this method can be useful for both diagnosis of psoriatic arthritis and evaluation of treatment efficacy through changes in nail characteristics. The review is interesting and well written. However, it needs some comments from the authors:
1) The authors should discuss the ideal probing frequency for studying psoriatic nails. In fact, they describe work in which different frequency probes (24 MHz or 27 MHz) have been used.
2) The advantage of studying the nail with ultrasound over that with dermoscopy should be clarified.
3) The review presented is narrative in nature. The authors should suggest a possible systematic review with meta-analysis or RCTs to reach more definitive conclusions on the clinical value of nail ultrasound.
4) The authors should also comment on how some drugs may be more effective on PsO than on PsA. In particular, they should clarify whether or not the improvement in psoriatic onychopathy observed with ultrasound after therapy is always accompanied by an improvement in the joint component in PsA.
5) It should be remembered that the diagnosis of psoriatic onychopathy sometimes requires biopsy examination. What is the sensitivity and specificity of studying the nail by ultrasound compared with biopsy as suggested by the reported studies?
English needs only minor editing
Author Response
R1.1) The authors should discuss the ideal probing frequency for studying psoriatic nails. In fact, they describe work in which different frequency probes (24 MHz or 27 MHz) have been used.
A1.1) Indeed, there is such variability in the literature. The following discussion has been added to the Introduction: “The lack of a standardized ultrasound imaging technique is apparent in the cited litera-ture as studies report using 15 MHz to 24 MHz ultrasound probes. This variability is mostly caused by cost and market availability, since 18-22 MHz probes are more widely available and purchased. Physically, the quality of nail strata images increases with probe frequency, therefore ideally 22-24 MHz should be used primarily in assessing nails in psoriasis and PsA.”
R1.2) The advantage of studying the nail with ultrasound over that with dermoscopy should be clarified.
A1.2) We thank the reviewer for pointing out an important aspect of the discussion. Accordingly, the article has been updated and the following text has been added to the Introduction: “Apart from clinical examination, the psoriatic nail can be evaluated by dermatoscopy and ultrasound. A 2022 systematic review on nail imaging techniques in psoriasis patients revealed that ultrasound is the most widely used technique for assessing psoriatic nails because, despite the fact that dermatoscopy has a lower cost and requires less training. The main advantage of ultrasound over dermatoscopy is the ability of the former to visualize DIP joints and distal insertions of the finger extensor tendons.”
R1.3) The review presented is narrative in nature. The authors should suggest a possible systematic review with meta-analysis or RCTs to reach more definitive conclusions on the clinical value of nail ultrasound.
A1.3) This is an important aspect which has been inserted in the Introduction, as follows: “PUBMED and Web of Science were searched using combinations of the following key-words: “ultrasound” and “nail” with either “psoriatic”, “psoriasis” or “DIP”, revealing 106 non-duplicate titles, of which 46 were relevant to the present review and 22 studies were referenced for their original research content. A systematic review with meta-analysis could not be performed on this study pool due to high design variability and few standard randomized intervention trials. Future development of the study niche could reveal me-ta-analytical content regarding ultrasound nail parameters and therapeutic nail response (see below).”
R1.4) The authors should also comment on how some drugs may be more effective on PsO than on PsA. In particular, they should clarify whether or not the improvement in psoriatic onychopathy observed with ultrasound after therapy is always accompanied by an improvement in the joint component in PsA.
A1.4) This is an interesting clinical observation and, upon the reviewer’s suggestion, it has been further underlined in the text regarding acitretin (“Of note, acitretin seems to be ineffective at the enthesis level and the improvement in pso-riatic onychopathy observed with ultrasound after acitretin therapy was not accompanied by an improvement in the joint component in PsA.”), methotrexate (“Unlike acitretin, methotrexate induced simultaneous ultrasound improvement in psoriatic onychopathy and PsA joint/enthesitis component.”) and secukinumab (“Subcutaneous secukinumab in PsA patients [30] also reduced mNAPSI, a slower radio-logical progression being noticed in patients with concomitant nail psoriasis.”).
R1.5) It should be remembered that the diagnosis of psoriatic onychopathy sometimes requires biopsy examination. What is the sensitivity and specificity of studying the nail by ultrasound compared with biopsy as suggested by the reported studies?
A1.5) We appreciate this essential discussion point which is now further discussed in the text by adding the following paragraph: “Histological studies are lacking in this field in the last years. As a consequence, there are no estimates of sensitivity and specificity of nail ultrasound compared to nail biopsy which is the gold standard for diagnosis nail onychopathy in psoriasis/PsA. Nail biopsy is rarely mandatory, unless nail involvement is still unclear after non-invasive measures (clinical aspect, dermatoscopy, ultrasound). Histological evaluation of DIP and enthesis at this level is done by lateral and longitudinal biopsy with an indirect evaluation of the enthesis. The procedure requires expertise in order to avoid diagnostic errors.”
Reviewer 2 Report
The authors present a narrative review of the utility of ultrasonography of the nail in the evaluation of psoriatic disease. To do this, they analyze the literature of the last 5 years.
Some observations from this reviewer with a view to a general improvement of the manuscript and a more attractive reading:
- Please be more explicit about how you selected the articles for the review, for example, which manuscript repository did you use?...pubmed only?, others? If some articles were not included in your review, please explain why, did you review abstracts sent to international conferences such as EULAR, ACR, others? etc, etc.
- In the only figure of your manuscript, you explain very well what is seen, but you do not point out the changes in the image. Please note that many readers may not be familiar with this type of ultrasound image. Please, use arrows, asterisks, or other formats, to indicate the main changes that the reader should find in the provided image.
- It would be desirable to provide an image of the normal ultrasound structure of the nail and its annexed structures, clearly indicating all those relevant to understanding the pathological changes that you later mention in the text.
- The abbreviation NPT is not defined the first time it appears in the text. Please correct it.
- For a more comfortable reading, introduce tables with the changes of the Worstman classification. If possible, mark those changes on an ultrasound image.
- Summarize the various studies reviewed in your manuscript with a table that includes:
- author, year of publication, methods, main findings and conclusions, whenever possible.
- It would be desirable to read a personal opinion from the authors on the true value of carrying out this type of study in clinical practice: what do they really contribute? Is it worth implementing the ultrasound equipment with the necessary probes to carry out this type of study? to what extent can they guide treatment decisions? do you think this type of literature is biased by the contribution of studies with only positive results? etc, etc.
- Focus more on the weaknesses of the studies you have reviewed: what quality are they? have possible confounders been taken into account in terms of the ultrasound findings? (for example, has it been taken into account if these patients had heavy handwork, which could skew the results, etc?)
- Reference 38 referring to Piero Ruscetti is actually 39. Please review the entire text to avoid these inconsistencies.
- Finally, shed some light on the research agenda in this field: what remains to be known or demonstrated? Is there a new technique on the horizon in this field? What role can these findings play in therapeutic decision-making? Is it worth doing a clinical trial about it? etc
Some minor typo need to be corrected. Please, review throughout the text.
Author Response
R2.1. Please be more explicit about how you selected the articles for the review, for example, which manuscript repository did you use?...pubmed only?, others? If some articles were not included in your review, please explain why, did you review abstracts sent to international conferences such as EULAR, ACR, others? etc, etc.
A2.1. We thank the reviewer for this mandatory information. The introduction was updated with the following text: “PUBMED and Web of Science were searched using combinations of the following key-words: “ultrasound” and “nail” with either “psoriatic”, “psoriasis” or “DIP”, revealing 106 non-duplicate titles, of which 46 were relevant to the present review and 22 studies were referenced for their original research content. A systematic review with meta-analysis could not be performed on this study pool due to high design variability and few standard randomized intervention trials. Future development of the study niche could reveal me-ta-analytical content regarding ultrasound nail parameters and therapeutic nail response (see below).”
R2.2. In the only figure of your manuscript, you explain very well what is seen, but you do not point out the changes in the image. Please note that many readers may not be familiar with this type of ultrasound image. Please, use arrows, asterisks, or other formats, to indicate the main changes that the reader should find in the provided image.
A2.2. Indeed, the ultrasound image should speak for itself. We have made the necessary marks on the image and we have re-inserted it into the text.
R2.3. It would be desirable to provide an image of the normal ultrasound structure of the nail and its annexed structures, clearly indicating all those relevant to understanding the pathological changes that you later mention in the text.
A2.3. A second figure was added to the manuscript, showing a normal nail with structure indications.
R2.4. The abbreviation NPT is not defined the first time it appears in the text. Please correct it.
A2.4. Done.
R2.5. For a more comfortable reading, introduce tables with the changes of the Worstman classification. If possible, mark those changes on an ultrasound image.
A2.5. A table of Worstman classification was added to the manuscript.
R2.6. Summarize the various studies reviewed in your manuscript with a table that includes: author, year of publication, methods, main findings and conclusions, whenever possible.
A2.6. A table summarizing these pieces of information was added to the manuscript.
R2.7. It would be desirable to read a personal opinion from the authors on the true value of carrying out this type of study in clinical practice: what do they really contribute? Is it worth implementing the ultrasound equipment with the necessary probes to carry out this type of study? To what extent can they guide treatment decisions? Do you think this type of literature is biased by the contribution of studies with only positive results? etc, etc.
A2.7. These are indeed useful reading highlights which have all been addressed in the manuscript by adding the following text: “Clinical practice can benefit from such studies. For patients with nail psoriasis and DIP pain without joint swelling, ultrasound detects subclinical changes of PsA (which allows treatment adjustment) and differentiate them from concomitant osteoarthritis. For the most common measurements, the 18 MHz probe is frequently found in ultrasound laboratories and no further hardware requirements are needed. Therefore, monitoring nail ultrasonographic parameters is readily available and it can be a useful tool in monitoring PsA treatment, as Nash et al. [31] suggested in their secukinumab study. The advent of new biological and targeted synthetic drugs that can specifically target GRAPPA domains of PsA (the nail and the enthesis) could imply that the evaluation of their effectiveness should not be based only on clinical examination, but should also include precisely quantified imaging methods. The literature offers studies that have proven their ability to modify ultrasonographic parameters on nails and entheses or only on nail parameters. The literature also offers mixt, balanced results, with both positive and negative reports regarding morphology and response to treatment, which leaves room for bigger and more controlled studies.”
R2.8. Focus more on the weaknesses of the studies you have reviewed: what quality are they? have possible confounders been taken into account in terms of the ultrasound findings? (for example, has it been taken into account if these patients had heavy handwork, which could skew the results, etc?)
A2.8. A statement on this topic has been added to the manuscript: “Generally, the weaknesses of the presented studies include insufficient data on confounders (e.g., professions involving repetitive and/or intense hand movement), small number of patients enrolled in single centers, the lack of control subgroups in therapeutic studies (with few exceptions).”
R2.9. Reference 38 referring to Piero Ruscetti is actually 39. Please review the entire text to avoid these inconsistencies.
A2.9. The entire text was revised and references were re-numbered correctly.
R2.10. Finally, shed some light on the research agenda in this field: what remains to be known or demonstrated? Is there a new technique on the horizon in this field? What role can these findings play in therapeutic decision-making? Is it worth doing a clinical trial about it? Etc.
A2.10. A statement on these topics has been added to the manuscript: “Certainly, ultrasonography and histology studies will be able to demonstrate the sensitivity and specificity of the imaging method. Ultrasound can also become the main imaging tool for dermatology nail evaluation, both for inflammatory conditions such as psoriasis and PsA, but also to make the differential diagnosis with certain benign or malignant tumours or with various subungual collections. Ultrasound can also guide nail biopsy. The methodology for quantifying the Doppler signal at the periungual level in PsA can also enter the research agenda, considering that it has variability in healthy controls. Even though we dispose of a core domain set published by GRAPPA that includes entheses and nails [42], we do not yet have standardized methods for measuring the nail parameters and the distal entheses. The development of these standards would make it easier to compare therapeutic studies on the core-set domains.”
Round 2
Reviewer 2 Report
Dear Authors
Your manuscript looks much better now and is more comprehensive.
Congratulations